# Science diplomacy and the 5th International Polar Year (IPY-5): planetary considerations across centuries

Paul Arthur Berkman[1,2,3,4,5]

[1]Science Diplomacy Center™, Falmouth, MA, USA; [2]International Science Council, Paris, France; [3]Program on Negotiation, Harvard Law School, Cambridge, MA, USA; [4]International Institute of Science Diplomacy and Sustainability (IISDS), UCSI University, Kuala Lumpur, Malaysia and [5]United Nations Institute for Training and Research (UNITAR), Geneva, Switzerland

## Perspective

**Keywords:**
Earth systems; resilience; transdisciplinary; sustainability; holistic; peace

**Corresponding author:**
Paul Arthur Berkman;
Emails: pab@scidiplo.org; pberkman@law.harvard.edu; paul.berkman@unitar.org

## Abstract

The 5th International Polar Year (IPY-5) in 2032–2033 represents an important next step in the legacy of the oldest continuous climate research program created by humanity, which intentionally began during a Solar Maximum with IPY-1 in 1882–1883, following the Little Ice Age. Current IPY-5 planning by the International Arctic Science Committee (IASC) and Scientific Committee on Antarctic Research (SCAR) is "From IPY-4 to IPY-5" with scope since 2007–2008, considering relevant large-scale polar process, international activities and UN decades. Additionally, there are essential features to incorporate into IPY-5 planning with Indigenous knowledge as well as next-generation leadership along with international science connections across the United Nations, involving core integration of data system and Earth–Sun system research, which accelerated with the International Geophysical Year (IGY) in 1957–1958 that was renamed from IPY-3. As memorialized in the 1959 Antarctic Treaty: "the International Geophysical Year accords with the interests of science and the progress of all mankind." Importantly, at the height of the Cold War with "forever" legacy, the 1959 Antarctic Treaty became the first nuclear arms agreement, applying science diplomacy among allies and adversaries alike based on "matters of common interest." Recognizing current challenges to enable inclusive dialogues – especially in the Arctic – planning for IPY-5 is far enough into the future to be imaginative and hopeful but close enough to be practical, especially to produce synergistic outcomes that inspire and empower next-generation leaders across the International Decade of Sciences for Sustainable Development from 2024 to 2033. Planning "From IPY-3 to IPY-5" – this invited *Cambridge Prisms Perspective* extends and amplifies the IASC-SCAR concept with its visionary principles – "striving for holistic, systemic, transdisciplinary research approaches" – for the benefit of all on Earth across generations.

## Impact statement

This invited *Cambridge Prisms: Coastal Futures* Perspective provides a transdisciplinary roadmap for Earth system scientists and next-generation science diplomats to help plan as well as implement the 5th International Polar Year (IPY-5) in 2032–2033 with local-to-global considerations in view of our shared sustainable development on Earth. As an essential case-study for humanity to operate on a planetary scale across centuries – with historical context, the IPY experiment is the oldest continuous research program to study Earth's climate, starting with IPY-1 in 1882–1883 during a Solar Maximum after the Little Ice Age in Europe. Renamed from IPY-3 – the International Geophysical Year (IGY) in 1957–1957 also was conducted during a Solar Maximum, but with lessons at the heart of world peace – beyond shortsighted nationalistic considerations with conflicts to resolve – applying "matters of common interest" that involve our survival as a globally-interconnected civilization. The first satellites were launched during the IGY with insights that enabled superpower adversaries to operate together among 67 nations with the "interests of science and the progress of all mankind," laying the foundation for the 1959 Antarctic Treaty to become the first nuclear arms agreement. The short-to-long term implications of this article are envisioned to enhance science with society, revealing "transdisciplinary" synergies across the natural sciences, social sciences and Indigenous knowledge with momentum building across the International Decade of Sciences for Sustainable Development from 2024 to 2033. With hope as the antidote for fear in our world, addressing exponential impacts across security-to-sustainability time scales – this paper introduces the first International Century as a concept to awaken with IPY-5, applying science diplomacy for the benefit of all on Earth across generations. If we think it! We can build it!

## 5th International Polar Year (IPY-5)

The 5th International Polar Year (IPY-5) is being planned for 2032–2033 as a "crucial new phase in a 150-year-old process," currently building on contributions "From IPY-4 to IPY-5" (International Arctic Science Committee and Scientific Committee on Antarctic Research [IASC-SCAR] 2023, 2024). Implications of IPY-5 are far more consequential for humanity, however, beyond the 4th International Polar Year (IPY-4) in 2007–2008, extending to the International Geophysical Year (IGY) 1957–1958 that was renamed from the 3rd International Polar Year (IPY-3).

IPY-5 is a rare research opportunity, when there is heightened funding nationally and internationally for current and next-generation leaders to shine. IPY-5 also will coincide with culmination of the International Decade of Sciences for Sustainable Development (IDSSD) 2024–2033 (United Nations Educational, Scientific and Cultural Organization [UNESCO] 2024), awakening questions about global synergies to stimulate by enhancing international scientific cooperation across the coming decade (Figure 1) with "science as a global public good" (Boulton 2021).

Operating across generations is at the heart of sustainability. The challenge is short-to-long term to balance economic prosperity, environmental protection and societal well-being with lessons learned and applied throughout (Figure 1). In view of the *"150-year-old IPY process,"*:

- Should the IPY-5 concept be limited to IPY-4 lessons (IASC-SCAR 2023, 2024), appreciating there were profound IPY-4 contributions, especially with Indigenous rights and sovereignty (Inuit Circumpolar Council 2009) as well as next-generation leadership (Cheek and Baeseman 2009)?
- What are the IPY-5 contributions that will be most helpful for humanity across the 21st century?
- Can IPY-5 become a transformational moment in the 21st century?

Addressing these questions is the goal of this paper – to inspire and empower next-generation leaders – harmonizing with the *Initial Concept Notes* for IPY-5, which are guided by a broad set of principles: "striving for holistic, systemic, transdisciplinary research approaches" (IASC-SCAR 2023, 2024).

## Holistic integration for Arctic coastal-marine sustainability

This paper also is about science diplomacy as a "language of hope" (Berkman 2020a) to inspire next-generation leaders. Addressing the *Cambridge Prisms: Coastal Futures* audience – synergies are introduced with *Holistic Integration for Arctic Coastal-Marine Sustainability*, which was the sub-text of the intertwined *Arctic Options/Pan-Arctic Options* projects from 2013 to 2022 (Berkman et al. 2020). From the start of these integration projects, holistic was

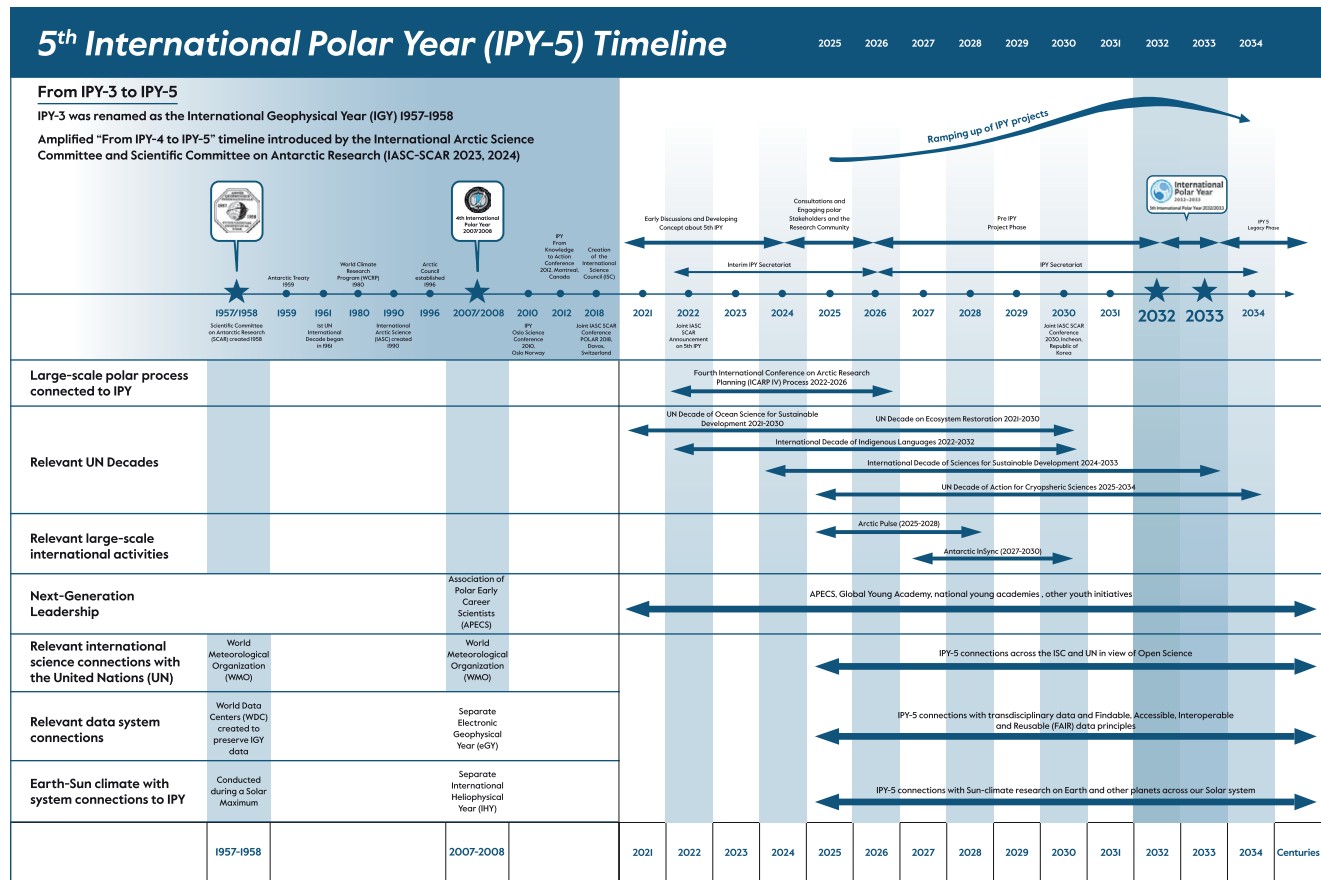

**Figure 1.** Amplified International Polar Year (IPY) planning "From IPY-3 to IPY-5," proposed herein to extend the timeline "From IPY-4 to IPY-5" that has been introduced by the International Arctic Science Committee and Scientific Committee on Antarctic Research (IASC-SCAR 2023, 2024). Planning with IPY-5 certainly will include relevant large-scale polar process, international activities and UN decades since IPY-4 in 2007–2008. Additionally, there are essential features to incorporate into IPY-5 planning with Indigenous knowledge as well as next-generation leadership along with international science connections across the United Nations (UNESCO 2021), involving core integration of data system and Earth–Sun system research, both of which accelerated with the International Geophysical Year (IGY) in 1957–1958 that was renamed from IPY-3.

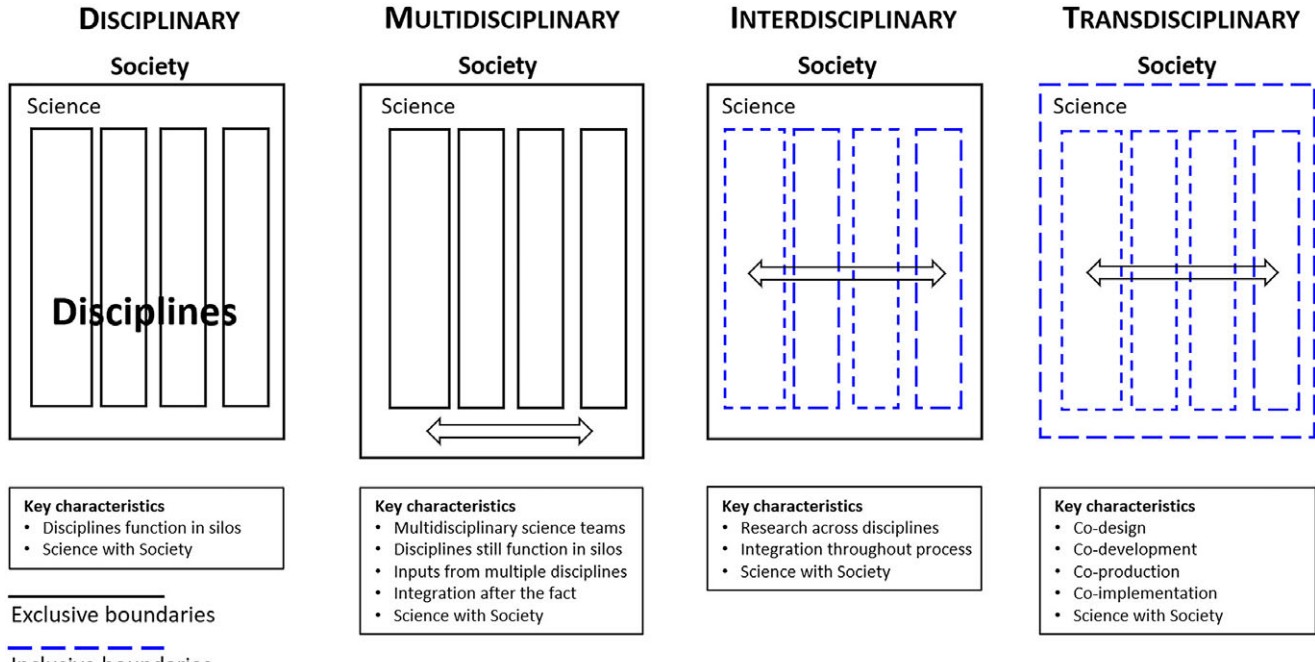

**Figure 2.** Evolution of science with society contributions, emphasizing disciplinary as a root concept with natural sciences, social sciences and Indigenous knowledge toward transdisciplinary integration as an aspiration with inclusion (who, what, when, where, why and how). Adapted from Takeuchi (2014).

defined as "international, interdisciplinary and inclusive," recognizing that inclusion is the singular challenge in the evolution of science–society relationships (Figure 2).

*Holistic integration for Arctic coastal-marine sustainability* involved questions to interpret the changing dynamics of biogeophysical and socioeconomic systems across the High North (Figure 3). *International* was represented with support from national funding agencies in Canada, China, France, Norway, Russia and the United States. *Interdisciplinary* was represented with natural scientists, social scientists and Indigenous knowledge holders all of whom reveal patterns, trends and processes (albeit with different methods), providing bases for decisionmaking. These knowledge systems, all of which have evolved over millennia – together with science as the "study of change" – highlight the challenge to be *inclusive*, integrating the six elements of discovery (who, what, when, where, why and how).

The ice is diminishing across thresholds in the Arctic with climate warming, where it is amplified four times that of the global average (Rantanen et al. 2022) – as a canary in the coal mine. Ice-climate feedbacks also are exacerbating the problem with diminishing planetary albedo (Winton 2006) in both polar regions and methane outgassing from the Arctic that is increasing greenhouse gases in Earth's atmosphere (Isaksen et al. 2011).

Symbolically, just one generation ago, the Arctic Ocean was characterized by persistent multi-year sea-ice coverage with stable floating "ice islands" inhabited for decades (Copland and Mueller 2017). In the Southern Ocean, by contrast, sea-ice was advancing and retreating annually across 3–21 million square kilometers (Zwally et al. 1983). With diminishing Arctic sea-ice observed since the satellite record began in 1978, today, we clearly see annual advance and retreat of sea ice across the Arctic Ocean (NSIDC 2024), as illustrated with the 2012 sea-ice minimum (Figure 3), revealing open water between the North Pacific and North Atlantic. A new sea-ice state also is emerging around Antarctica with

decreasing ice-extent being recorded since 2016 with the record minimum in 2023 (Purich and Doddridge 2023).

Systems are defined by their boundaries and the Arctic Ocean already has undergone a boundary change, like removing the ceiling of a room. Viewed variously from perspectives of diverse stakeholders, implications of the new Arctic Ocean involve inherent security risks of political, economic and societal instabilities that are immediate. Simultaneously, there are urgencies to continuously address across generations at sustainability time scales. At the levels of peoples, nations and the world – the challenge is to operate across a "continuum of urgencies" to make informed decisions from security-to-sustainability time scales, requiring science with diplomacy to negotiate short-to-long term for the benefit of society (Figures 2 and 4).

At personal levels, informed decisionmaking (Figure 4) is like driving a car, involving immediate risks to the left and right with red lights in front to navigate into the future while viewing past circumstances in the rear-view mirror.

## Earth's oldest climate experiment

IPY-5 is on the horizon next decade with climate context that goes back to IPY-1 in 1882–1883. By 1850, Europe was exiting the "Little Ice Age" that had lasted more than three centuries with vast glaciers extending through the Alps and negative societal impacts across the continent (Berkman 2010). Afterward, with the International Meteorological Congress in 1873, planning began for IPY-1 (Luedecke 2004) with the International Meteorological Organization (IMO) emerging in 1878 (Tannehill 1947; WMO 2024) as the first organization in the world to exchange weather information among nations. The IMO operated until 1951 when it was replaced by the World Meteorological Organization (WMO). With IPY-1 and planetary considerations in relation to the Sun during a Solar

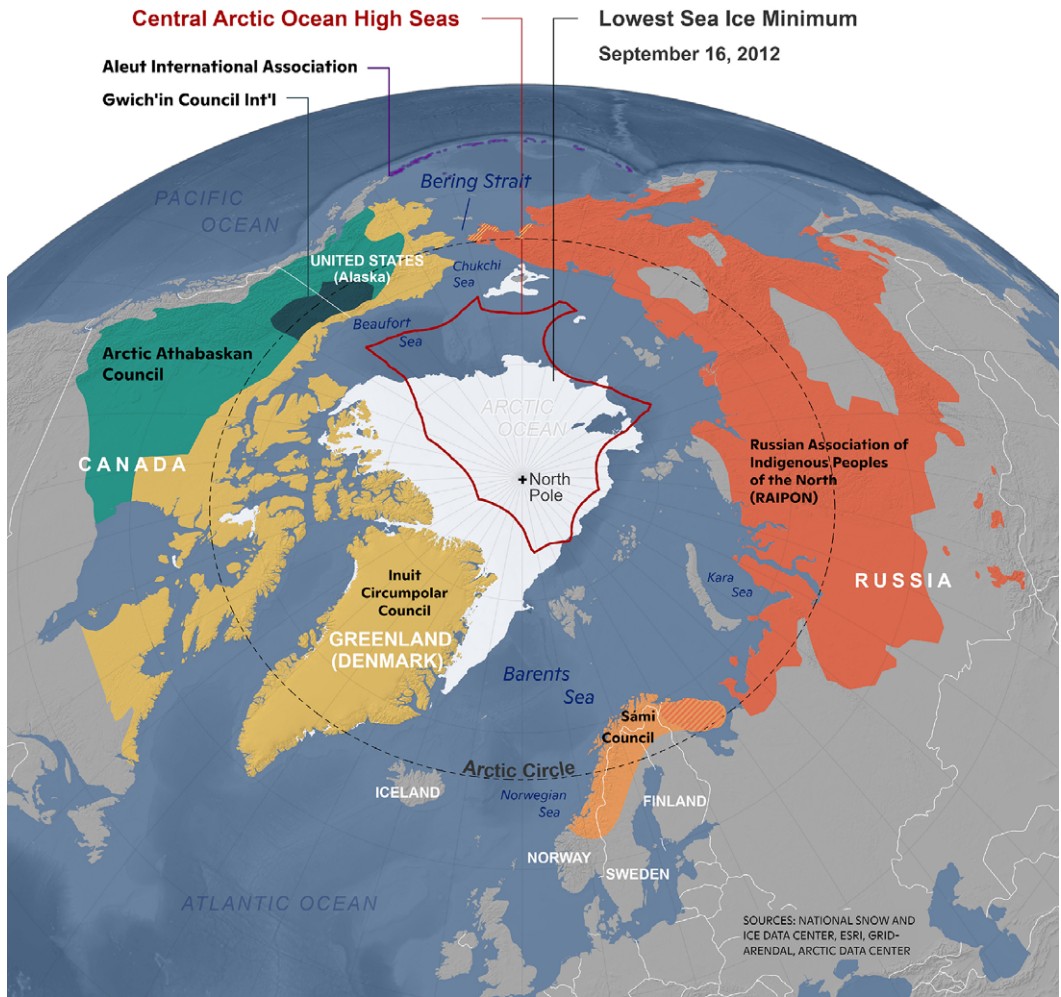

**Figure 3.** "Holistic, systemic, transdisciplinary" integration with the North Pole as a "Pole of Peace," applying Cold War lessons (Gorbachev 1987), with the eight Arctic states north of the Arctic Circle (Arctic Council 2024) and six Arctic Indigenous Peoples Organizations (IPS 2024). Biogeophysical features are illustrated with the 2012 sea-ice minimum (white area) in view of the Central Arctic Ocean (CAO) High Seas as an international space beyond sovereign jurisdictions (red boundaries). Color contrasting with names of Indigenous Peoples Organizations has been enhanced with this book-cover map from Berkman et al. (2022).

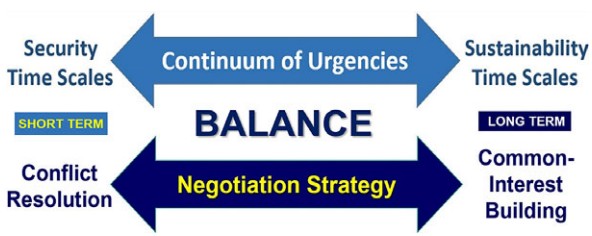

**Figure 4.** Informed decisionmaking – theoretical framework (see Figure 5) – as the "engine of science diplomacy" (Berkman 2020a). Elaborated from the Vienna Dialogue Team (2017), Berkman et al. (2022) and Council of Canadian Academies (2024).

Maximum (Table 1) – recognizing a polar connection with cold weather that had gripped Europe for centuries – the IMO was involved with launching the Earth's oldest climate experiment.

The IPY experiment with Earth's climate began in view of both polar regions (Barr and Luedecke 2010), which became the experimental control to interpret planetary forcing from the Sun, which is the primary external driver of climates on all celestial bodies in our Solar System. Sunspots had been studied for thousands of years, as reflected by Chinese parchments (Kirkwood 1869), representing

the path of accumulated knowledge discovery of humankind across the Earth, responding to the Sun and seasons. The timing of IPY-1 was specified to coincide with a Solar Maximum in the 11-year solar cycle of sunspots (Table 1), in contrast to IPY-2 during a Solar Minimum in 1932–1933.

The climate experiment was continued during another Solar Maximum with IPY-3, which was renamed the International Geophysical Year (IGY) in 1957–1958 – becoming the 20th-century threshold to study Earth's climate from the polar regions and globally with unrivaled international scientific cooperation on a planetary scale. However, the Solar context was missing with IPY-4 and instead there was a separate International Heliophysical Year in 2007. There also was a separate Electronic Geophysical Year in 2007–2008, coordinated by the Committee on Data (CODATA), World Data System (WDS) and other scientific unions through the International Council of Scientific Unions (ICSU). As a next step in the 150-year climate experiment (Table 1), IPY-5 will happen during mid-solar cycle, introducing potential synergies with Solar system observations that are being planned for 2032–2033 (Caspi et al. 2023).

A unifying feature of the IGY was groundbreaking technology with the first satellites, which transformed the cone of synoptic observations across the Earth's surface well beyond the scope of

**Table 1.** The International Polar Year (IPY) experiment

| Characteristics | IPY-1 1882–1883 | IPY-2 1932–1933 | IPY-3[a] 1957–1958 | IPY-4 2007–2008 | IPY-5 2032-2033 |
|---|---|---|---|---|---|
| Solar Activity | Solar Maximum | Solar Minimum | Solar Maximum | IHY[b] | Mid-Solar Cycle |
| Nations Participating | 11 | 40 | 67 | 60+ | to be determined |
| Disciplines | 3 | 4 | 14 | 11 | to be determined |
| Observation Distance from Earth | Ground-based | Balloon | Satellite | Satellite | **Space Vehicle** (proposed herein) |
| Geographic Focus | Arctic and Antarctic | Arctic and Antarctic | Earth | Arctic and Antarctic | **Earth** (proposed herein) |
| International Security Issue | 'Weather' | Radio | Satellites[c] | Polar | **Global** (proposed herein) |

Source: Adapted from Berkman (2003, 2020b)
[a] Renamed the International Geophysical Year (IGY)
[b] Solar-terrestrial focus was part of the International Heliophysical Year (IHY) in 2007
[c] First International Decade was introduced in 1961, following the IGY

previous IPY (Table 1). With these "scientific satellites," humankind leapt into outer space, as reflected by the first national space policies (Berkman 2011). Rocket systems to launch the "scientific satellites" were the same as those that subsequently enabled ballistic missiles during the Cold War. The period of the 1950s is similar to the world today when there is severe distrust, heightened animosity and minimized dialogue among superpowers, which were new in our globally interconnected civilization after the Second World War.

With satellites, the IGY reached into international security issues, which is an observation that can be extended across the IPY experiment (Table 1), with societal benefits at each stage (Figure 2): IPY-1 with weather, IPY-2 with the communication advance of radio, and IPY-4 with its polar focus. The IGY became a ray of hope in the darkness of Mutually Assured Destruction discussions. Enabled largely by ICSU and SCAR with connections to national academies inclusively around the world – the IGY contributed to global peace. The immediate outcome of the IGY was the Antarctic Treaty (1959) signed in Washington, DC, acknowledging the: "International Geophysical Year accords with the interests of science and the progress of all mankind." As recognized further in the Preamble of the 1959 Antarctic Treaty: "… it is in the interest of all mankind that Antarctica shall continue forever to be used exclusively for peaceful purposes."

- What enabled the 1959 Antarctic Treaty to become the first nuclear arms agreement?
- How did the 1959 Antarctic Treaty become the template for the 1967 *Treaty on Principles Governing the Activities of States in the Exploration and Use of Outer Space, including the Moon and Other Celestial Bodies*?
- Why did the United States and Soviet Union cooperate in Antarctica and Outer Space throughout the Cold War, despite animosities that isolated these superpowers elsewhere?

These inclusive questions are at the core of science diplomacy, learning and applying "forever" lessons with the 1959 *Antarctic Treaty*, as reflected by *Science into Policy: Global Lessons from Antarctica* (Berkman 2002). The subsequent Antarctic Treaty

Summit (2009) at the Smithsonian Institution in Washington, DC, which was an IPY-4 project, generated the first book on *Science Diplomacy* (Berkman et al. 2011): "For the benefit of present and future generations – the global challenge is to balance national interests and common interests. Science diplomacy is the international, interdisciplinary and inclusive process to achieve this global balance for the benefit of all life on Earth."

The 2009 Antarctic Treaty Summit also contributed to the 2009 *New Frontiers in Science Diplomacy* conference convened by The Royal Society (2010) at Wilton Park in the United Kingdom with the American Association for the Advancement of Science (AAAS), which awakened foreign ministries around the world to consider what is science diplomacy. From the pinnacle of foreign ministries across society – science diplomacy has become a field of study in its own right with diverse initiatives, programs, institutes and responsibilities at local-to-global levels. The "forever" challenge shared among all 8 billion of us remains "to balance national interests and common interests," as underscored over the past 150 years by Earth's oldest continuous climate experiment (Table 1).

### Informed decisionmaking with the future of humanity

Introducing the concept of an "international, interdisciplinary and inclusive process" begs the question: how does science diplomacy operate? The answer, in part, is revealed with lessons learned from the 2009 Antarctic Treaty Summit that were applied in 2010 to produce the first formal dialogue between the North Atlantic Treaty Organization and Russia regarding security in the Arctic (Berkman and Vylegzhanin 2013). What skills and methods enabled science diplomats from the outside, without the imprimatur of governmental authority (Gluckman et al. 2017), to engage superpower adversaries in such a dialogue? This question was a key focus with the *Arctic Options /Pan-Arctic Options* projects (Figure 3), operating across a "continuum of urgencies" (Figure 4), by serendipity during the threshold decade when five binding Arctic agreements entered into force among the eight Arctic states (Figure 3 and Table 2).

**Table 2.** Circumpolar complex of Arctic governance mechanisms after IPY-4

| LEGAL AGREEMENT TITLE | DATE | | SIGNATORIES |
|---|---|---|---|
| | Signed | In Force | |
| Agreement on Cooperation on Aeronautical and Maritime Search and Rescue in the Arctic | May 12, 2011 | January 19, 2013 | Eight Arctic States (Arctic-8): Canada, Denmark, Finland, Iceland, Sweden, Norway, Russia, United States |
| Agreement on Cooperation on Marine Oil Pollution Preparedness and Response in the Arctic | May 15, 2013 | March 25, 2016 | Arctic-8 |
| Agreement on Enhancing International Arctic Scientific Cooperation | May 11, 2017 | May 23, 2018 | Arctic-8 |
| International Code for Ships Operating in Polar Waters (Polar Code) | appending Conventions | January 1, 2017 | International Maritime Organization signatories through the United Nations |
| Agreement to Prevent Unregulated High Seas Fisheries in the Central Arctic Ocean | October 3, 2018 | June 25, 2021 | Canada, China, Denmark, European Union, Iceland, Japan, Korea, Norway, Russia, United States |

In particular, the 2017 *Agreement on Enhancing International Arctic Scientific Cooperation Arctic* accentuates cross-cutting responsibilities of science diplomats to enhance as well as protect "international scientific cooperation," which is among the "matters of common interest" memorialized with the "interests of all mankind" in the 1959 Antarctic Treaty one decade after World War II. This insight was a convergence (Berkman et al. 2017), revealing that science diplomats broker dialogues among allies and adversaries alike simply by introducing questions (Figure 5) rather than seeking answers or making recommendations. Being able to frame the questions with inclusion (who, what, when, where, why and how) is the skill.

Questions are the common foundation for research-into-action to produce informed decisions (Figure 5): neither good nor bad, right nor wrong, but decisions that optimize the available information inclusively, as the holistic process. Questions are the least complicated stage of engagement and lowest hanging fruit to operate with continuity across a "continuum of urgencies" short-to-long term (Figure 4), which is across decades-to-centuries in the context of Earth's climate (Table 1). The reality check is implementing the 1992 *United Nations Framework Convention on Climate Change* (UNFCCC), which is a "forever" challenge, requiring continuous Conferences of Parties into the 22nd century and beyond.

Questions enable triangulation with education, research and leadership, underscoring the holistic process with lifelong learning to build sustainable solutions for the world we live in (Figure 5). With Open Science (UNESCO 2021) – evolving with global inclusion independent of geopolitics – the natural sciences, social sciences and Indigenous knowledge together (Figure 2) offer humanity hope to address changes continuously across the spectrum of subnational–national–international jurisdictions (Berkman et al. 2019).

In the spirit of introducing options (without advocacy), which can be used or ignored explicitly – with respect to the decision-makers – integration of research-into-action (Figure 5) for the benefit of society (Figure 2) could be an explicit objective of IPY-5. The implications of IPY-5 are much larger than the IPY-4 scope (IASC-SCAR 2023, 2024) that currently is stimulating research funding nationally and internationally. Could IPY-5 accomplish for the 21st century, what IPY-3 achieved last century "with the interests of science and the progress of all mankind"?

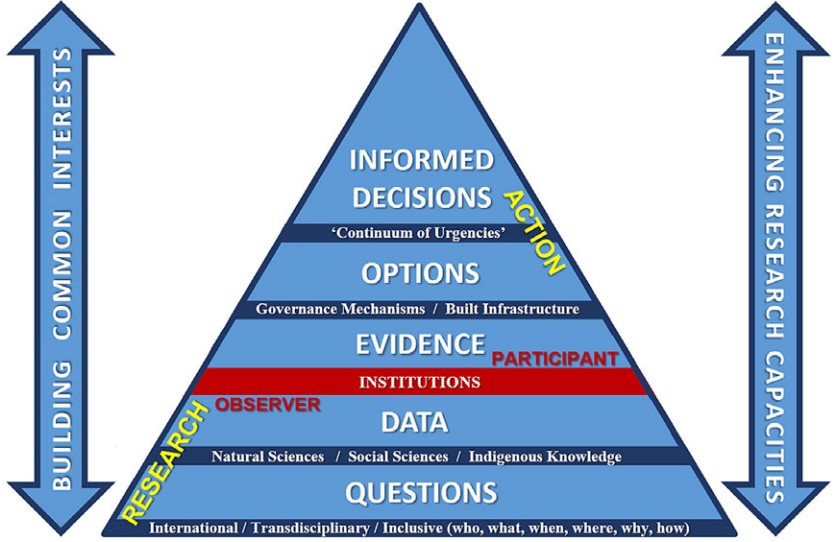

**Figure 5.** Informed decisionmaking – methodology framework (see Figure 4). Elaborated from Berkman et al. (2017, 2020, 2022).

## Reaching to the stars

Extrapolating from ground level in 1882–1883 to balloons in 1932–1933 to satellites in 1957–1958 (Table 1), placed potential Earth observations somewhere among the other planets with IPY-4 (Berkman 2003). Distant observations of climate dynamics on planets and celestial bodies across our Solar System (National Academies of Sciences, Engineering, and Medicine 2023) remain as a potential opportunity with IPY-5 to understand our home planet in the broader context, contributing to Earths' oldest climate experiment. Moreover – looking across the 21st century (UNEP 2024) – the path for humankind is increasingly into Outer Space with the Moon and other celestial bodies.

- What synergies are possible with synoptic Earth System and Solar System observations during IPY-5?

For example, early research with Earth's magnetic field lines at polar conjugate points was central to the IGY and discovery of the Van Allen radiation belts, noting James Van Allen along with Lloyd Berkner and Sydney Chapman proposed IPY-3 to become the IGY (Korsmo 2007). IPY-5 experiments across the Solar System could help to understand "space weather" from the Sun, which has economic impacts with critical infrastructure annually in the billions and perhaps trillions of dollars (Schulte in den Bäumen et al. 2014). More broadly, Solar System perspectives open the imagination to see the Earth System with transdisciplinary (Figure 2) vantages across the natural sciences, social sciences and Indigenous knowledge.

Modeling complexities of the Earth System for societal benefit is an ongoing journey (Figure 6), which is being accomplished iteratively with increasing global inclusion across the IPY experiment (Table 1). This proposition can be tested by considering international initiatives to operate progressively over longer periods (Figure 4), imagining back to the late 19th century when the first IPY launched global science.

In view of global science, operating short to long term (Figure 4), another threshold was traversed after the IGY with the first International Decade in 1961 (UNESCO 1961, Figure 7). Among the relevant decades (Figure 1), planning across IDSSD 2024–2033 directly complements the timing and scope of IPY-5 as well as the cross-cutting societal contribution of the IPY experiment (Table 1):

- How will IPY-5 enhance the integration of Earth System perspectives across the natural sciences, social sciences and Indigenous knowledge on a planetary scale, including with the Third Pole (Yao et al. 2012) as well as other cryospheric regions?
- Like its IPY predecessors from the Cold War forward, how can IPY-5 become a catalyst to enhance "international scientific cooperation" among superpowers, especially now in the warming Arctic (Figure 3) as well as in the Antarctic, where national tensions are heating to compromise the Antarctic Treaty System?
- How will IPY-5 integrate with and enhance other International Decades, such as the Decade of Action for Cryospheric Sciences 2025–2034 (UNESCO 2025), in view research-into-action (Figure 5) with societal benefits from transdisciplinary insights (Figure 2)?

Planning for IPY-5 is far enough into the future to be imaginative and hopeful but close enough to be practical across our globally-interconnected civilization. Safer drivers look further down the road – maneuvering with informed decisions (Figure 5)

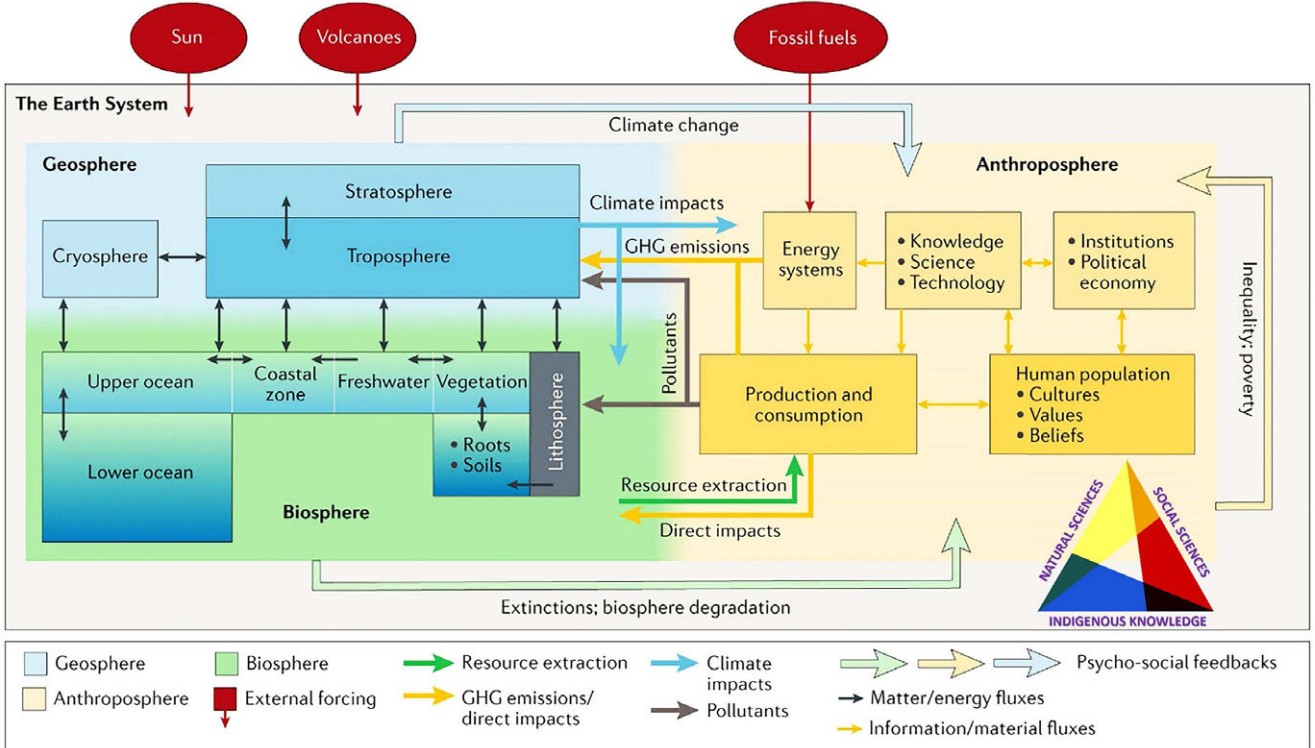

**Figure 6.** Updated conceptual model of the Earth system (Steffen et al. 2020), evolving from the Bretherton (1985) diagram. Triangulation of natural sciences, social sciences and Indigenous knowledge is added to inspire synergies across the spectrum of subnational-national-international jurisdictions (Berkman et al. 2022) with progress across generations for our shared sustainable development as a globally-interconnected civilization.

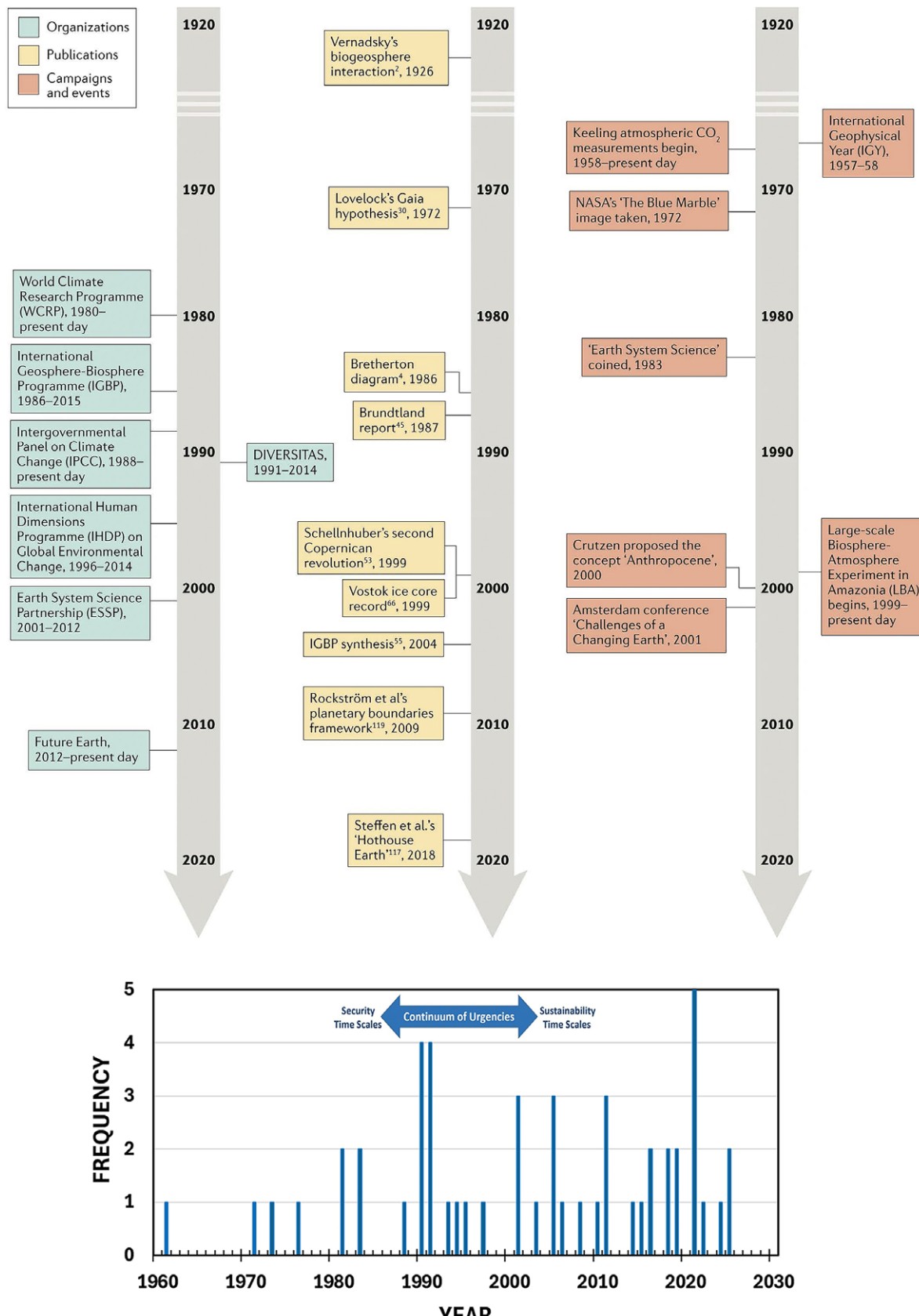

**Figure 7.** Earth system science after the International Geophysical Year (IGY) 1957–1958, which was renamed from the 3rd International Polar Year (IPY-3). Qualitative perspectives from Steffen et al. (2020) in view of organizations, publications and events (upper); quantitative perspectives across a 'continuum of urgencies' (Figure 4) in view of the frequency of United Nations International Decades that emerged in 1961 (lower), utilizing data from United Nations (2024a).

in view of red lights and traffic ahead as well as upcoming from the rear. Looking further down the road is coupled with science-society relationships over time (Figure 2), from security-to-sustainability time scales (Figure 4). From our perspective today in the 21st century – in view of the IPY experiment that began in the 19th century – humanity can be seen to be operating with continuity on a planetary scale across centuries (Table 1).

Reaching across longer periods on a planetary scale, an informed decisionmaking (Figures 4 and 5) threshold with science-society relationship was the first International Decade (Figure 7), beyond the duration of International Years, Weeks and Days after the IGY (United Nations 2024a, 2024b, 2024c). Interpreted further, the frequency of International Decades has increased from 1961 to the present, during and after the Cold War. Figure 7 further reveals a global threshold with common-interest building after 1991.

It is perhaps without surprise that the *United Nations Conference on Environment and Development* (UNCED) happened in Rio de Janeiro in 1992 immediately following the Cold War – when international dialogues were enhanced inclusively. Planning had been underway beforehand to produce a *"global agenda for change,"* which was the task of the World Commission on Environment and Development, as reported in *Our Common Future* in 1987 (WCED 1987). The Cold War ending in 1991 was the inflection point, followed by the 1992 Rio Conference, which enabled the UNFCCC as well as the Convention on Biological Diversity (CBD) and United Nations Convention to Combat Desertification (UNCCD).

Operating before-through-after an inflection point (Berkman 2020c) – the three Rio conventions illustrate the outcome of informed decisionmaking as a holistic process, enabled by building common interests short-to-long term (Figures 4 and 5). Analogous outcomes of informed decisionmaking are illustrated with the September 1945 inflection of the Second World War end. Beforehand, there was framing for the global order with the Food Agriculture Organization (FAO) in 1943 (United Nations 1943) and international monetary system at Bretton Woods in 1944 (Bordo 1993), leading to the San Francisco conference in April 1945 with the *Charter of the United Nations and Statute of the International Court of Justice* (United Nations 1945).

- Could the end of wars in Ukraine or the Middle East or elsewhere (Geneva Academy 2025) be global inflection points?
- Could IPY-5 with Earth System considerations (Figure 6) and common-interest building (Figures 4 and 5) help humanity to operate before-through-after global inflection points now and across the 21st century?

## IPY-5 as a global threshold

As with ICSU coordination of IPY-3 and IPY-4 – international coordination of IPY-5 will require leadership with the International Science Council (ISC), creating synergies among its many unions and committees (including CODATA, IASC, SCAR and WDS) in connection with national academies, science foundations and related institutions, ultimately engaging science pioneers. Moreover, ISC partnerships with IPY-5 involve United Nations organizations, including the UNESCO that manages the International Decades (Figure 7) as well as the WMO, which is successor to the IMO during the first two IPY. As stated by Celese Saulo (2024), current WMO Secretary-General: "Let us come together with one vision and one goal: to protect people, livelihoods and the future. Early warnings work. They must work for everyone."

Additionally, IPY-5 will involve the interplay of international institutions established under the Charter of the United Nations, including the Antarctic Treaty, UNFCCC, CBD, UNCCD and those identified in Table 2. In turn, diverse nongovernmental organizations at national and international levels along with rightsholders and stakeholders across society inclusively (Figure 3) are involved. The institutional interplay with IPY-5 will involve research-into-action (Figure 5), which is a two-way street to implement this next step in the oldest continuous climate experiment created by humanity (Table 1), responding to and influencing local-to-global affairs.

Importantly, there is leadership with Indigenous Peoples in the Arctic (Aleut International Association, Arctic Athabaskan Council, Gwich'in Council International, Inuit Circumpolar Council, Russian Association of Indigenous Peoples of the North (RAIPON) and Saami Council), who have been resilient over millennia (Inuit Circumpolar Council 2009) to survive in the face of extreme environmental and ecosystem changes connected with Earth's climate. There are "forever" lessons with Indigenous cultures to operate across "planetary boundaries" (Richardson et al. 2023; Rockström et al. 2024). It is hopeful for humanity that Indigenous youth are insisting to be at the forefront of climate diplomacy (Sogbanmu et al. 2023).

We still are in the preparatory-planning phase of IPY-5, before the project phase begins in 2026, as introduced by IASC-SCAR (2023, 2024), with consideration of the legacy afterward across centuries (Figure 1). The challenge with IPY-5 is to be holistic with local-to-global considerations and transdisciplinary (Figure 2) capacities, addressing questions of common concern across the Earth (Figure 5), as embodied in the United Nations Sustainable Development Goals (United Nations 2015).

- As low-hanging fruit with egalitarian opportunities en route to IPY-5 – operating across a "continuum of urgencies" (Figure 4) – how are the International Days, Weeks, Years and Decades (United Nations 2024a, 2024b, 2024c) synergistic in ways that will help to inform (Figure 5) as well as reinforce progress with our sustainable development at local-to-global levels?

It is noteworthy that the United Nations observed the first International Year in 1959 (United Nations 2024b) as well as the first International Decade in 1961 (United Nations 2024c; Figure 7), following the IGY in 1957–1958.

Goal of this paper is to inspire and empower next-generation science diplomats to build IPY-5 with considerations across the 21st century, anticipating many of you will be living into the 22nd century. Human lifespans represent a key challenge of the Anthropocene (Crutzen 2006), now to operate across centuries, which is the short-to-long term period seen also in the rear view across the IPY experiment from the 19th century forward (Table 1).

The IGY in 1957–1958 accelerated Earth system science (Figures 6 and 7) and explicitly stimulated the 1959 Antarctic Treaty "with the interests of science and the progress of all mankind." Building common interests with transdisciplinary imagination and inclusion "From IPY-3 to IPY-5" (Figures 1–7 and Tables 1 and 2):

**Will IPY-5 awaken the first International Century among its legacies with science diplomacy to transform research-into-action for the benefit of all on Earth across generations?**

**Open peer review.** To view the open peer review materials for this article, please visit http://doi.org/10.1017/cft.2025.2.

**Acknowledgments.** I thank Saleem Ali for the opportunity to share observations in a panel dialogue *about Arctic Science Diplomacy: Charting a Path Through Conflict* at the 2024 American Association for the Advancement of Science (AAAS) annual meeting, which awakened the invitation from *Cambridge Prisms* to craft this *Perspective*. I thank Alik Ismail-Zadeh with the International Science Council (ISC), Yeadong Kim with the Scientific Committee on Antarctic Research (SCAR) and Henry Burgess with the International Arctic Science Committee (IASC) for sharing their helpful feedback on the initial draft of this manuscript before it was submitted. I thank Greg Fiske with the Woodwell Climate Research Center for his cartographic assistance with Figure 3. I appreciate the thoughtful comments from two anonymous reviewers, who recommended publication with "minor" but important changes to improve this manuscript.

**Competing interest.** The author has no competing interests with this manuscript.

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
