## [Reviewer Report]

As it is the first time I review a Cambridge Prisms journal, I am not quite sure whether to include both ‘the minor changes’ and ‘my review’ here but cannot find any other place to place the two parts:

MINOR CHANGES AND REMARKS:

• The abstract was not – as mentioned in the invitation mail – ‘included at the bottom of this email’ and this is explained at the first page of the manuscript: Graphical abstract not yet produced!

• Page 1: Figure 1: is very illustrative but hard to read without a magnifier and must be made easier readable.

• Page 3: Figure 3: all names of the associations of the Indigenous Peoples must be made easier readable.

• List of figures and tables would be useful.

• List of abbreviations would be helpful.

• Page 11:33: an explicit mentioning of the Indigenous Peoples (referred to in Figure 3) would be useful.

REVIEW

The manuscript presents the historical background, the 150-year-old IPY process, for the International Polar Year, IPY-5, planed for 2032-2033, and highlights both the IPYs historical contexts and their contributions to “the progress of all mankind” facilitated by science diplomacy.

Based on ‘the IPY process’ as a learning experience and stressing results such as United Nations and other international organisations and conventions as “outcome of informed decisionmaking as a holistic process, building common interests short-to-long term”, the manuscript raises key questions and challenges to the IPY-5.

Among the challenges the manuscript stresses, is being ”inclusive (who, what, where, how and why) with local-to-global considerations and transdisciplinary capacities building questions of common concerns across the Earth”.

Not least for the people including the Indigenous Peoples living in the Arctic, the focus on transdisciplinarity with co-development, co-production and co-implementation and including Indigenous knowledge together with natural sciences social sciences, is important.

The manuscript is both a timely and an excellent contribution to the pre-planning process of IPY-5 and offers key questions as well as visions that will hopefully impact the project phase of the IPY-5 also.

---

## [Reviewer Report]

The paper ‘SCIENCE DIPLOMACY AND THE 5TH INTERNATIONAL POLAR YEAR (IPY-5): PLANETARY CONSIDERATIONS ACROSS CENTURIES’ is logically well-written and original. In the preparation of the 5th IPY, which SCAR and IASC are currently planning, I think it is a meaningful article that emphasises the importance of initial conceptualisation, meaning and social connectivity through science diplomacy. In particular, it is a very thoughtful proposal on how the 5th IPY can go beyond the accumulation of scientific knowledge on climate change and develop into an international agreement on polar regions, just as the IGY led to the Antarctic Treaty.

It is recommended for acceptance with a minor revision. Comments and suggestions are marked with balloons in the manuscript attached.

---

## [Editor Report]

Dear Paul

I am pleased to be able to tell you that we have now received the necessary reviews to your manuscript “SCIENCE DIPLOMACY AND THE 5TH INTERNATIONAL POLAR YEAR (IPY-5): PLANETARY CONSIDERATIONS ACROSS CENTURIES”. Both reviewers have recommended the manuscript be accepted subject to some very minor revisions that they have indicated. These revisions seem appropriate to me to clarify a few aspects of the text. I also concur with the reviewers comments regarding the readability of text in Figures 1 and 3 and ask you to see if the text can be made more readable - for Figure 3 I think this can be achieved simply by changing the colour of the text so it contrasts more against the background colours and for Figure 1 even a point size increase in text size might suffice without affecting the position of text against other graphics.

I am looking forward to seeing this article in print as it is indeed timely and an important contribution to the science-policy nexus for the Arctic.

Regards, Martin

---

## [Reviewer Report]

Thank you for addressing the review comments and for your contribution that is an excellent point of departure for the understanding of the IPY proces and the discussions and planning of IPY-5.

---

## [Reviewer Report]

My points were well taken. But I think there is still one part that needs more clarity.

The Antarctic Treaty was not established under the Charter of the United Nations. The Antarctic Treaty is an independent international agreement that came into force in 1961 and is not formally associated with the United Nations system, although it shares common goals with UN principles, such as promoting peace and scientific cooperation. However, on page 13, lines 37-39, it does not interpret that way to me.

If I suggest more accurate phrasing for this,

“IPY-5 will involve the interplay of international institutions established under the Charter of the United Nations, along with other key agreements such as the Antarctic Treaty, the UNFCCC, CBD, UNCCD and those….”

---

## [Editor Report]

Thank you for making the adjsutments to the paper in line with reviewer comments and for clarifying particular points with the Journal editors directly. I look forward to seeing the paper published and expect that it will have a very timely impact.

Kind regards, Martin